# ATF6 Activation Reduces Amyloidogenic Transthyretin Secretion through Increased Interactions with Endoplasmic Reticulum Proteostasis Factors

**DOI:** 10.3390/cells11101661

**Published:** 2022-05-17

**Authors:** Jaleh S. Mesgarzadeh, Isabelle C. Romine, Ethan M. Smith-Cohen, Julia M. D. Grandjean, Jeffery W. Kelly, Joseph C. Genereux, R. Luke Wiseman

**Affiliations:** 1Department of Molecular Medicine, The Scripps Research Institute, La Jolla, CA 92037, USA; 2Department of Chemistry, The Scripps Research Institute, La Jolla, CA 92037, USA; 3The Skaggs Institute for Chemical Biology, The Scripps Research Institute, La Jolla, CA 92037, USA; 4Department of Chemistry, University of California, Riverside, Riverside, CA 92521, USA

**Keywords:** ER proteostasis, protein aggregation, amyloid disease, unfolded protein response (UPR), ATF6, protein disulfide isomerase (PDI), extracellular proteostasis

## Abstract

The extracellular aggregation of destabilized transthyretin (TTR) variants is implicated in the onset and pathogenesis of familial TTR-related amyloid diseases. One strategy to reduce the toxic, extracellular aggregation of TTR is to decrease the population of aggregation-prone proteins secreted from mammalian cells. The stress-independent activation of the unfolded protein response (UPR)-associated transcription factor ATF6 preferentially decreases the secretion and subsequent aggregation of destabilized, aggregation-prone TTR variants. However, the mechanism of this reduced secretion was previously undefined. Here, we implement a mass-spectrometry-based interactomics approach to identify endoplasmic reticulum (ER) proteostasis factors involved in ATF6-dependent reductions in destabilized TTR secretion. We show that ATF6 activation reduces amyloidogenic TTR secretion and subsequent aggregation through a mechanism involving ER retention that is mediated by increased interactions with ATF6-regulated ER proteostasis factors including BiP and PDIA4. Intriguingly, the PDIA4-dependent retention of TTR is independent of both the single TTR cysteine residue and the redox activity of PDIA4, indicating that PDIA4 retains destabilized TTR in the ER through a redox-independent mechanism. Our results define a mechanistic basis to explain the ATF6 activation-dependent reduction in destabilized, amyloidogenic TTR secretion that could be therapeutically accessed to improve treatments of TTR-related amyloid diseases.

## 1. Introduction

Transthyretin (TTR) is a secretory protein that is primarily synthesized in the endoplasmic reticulum (ER) of the liver and secreted into the blood, where it circulates as a stable tetrameric protein that transports holo-retinol binding protein (RBP) and small molecules such as thyroxine [1,2]. The serum accumulation of TTR as soluble oligomers or amyloid fibrils is implicated in the pathogenesis of various systemic amyloid diseases [3,4]. Hereditary TTR Amyloidosis is a systemic amyloid disease that is largely dictated by the presence of inherited autosomal dominant single nucleotide polymorphisms in the *TTR* gene—of which there are more than 100 known mutants [5]—and can present with multi-tissue etiologies including cardiomyopathy and peripheral neuropathies, as well as multi-organ system etiologies [3,4]. Additionally, wild-type TTR can aggregate in the disease senile systemic amyloidosis, causing cardiomyopathy due to aggregates in the heart [3,4]. Numerous therapeutic strategies have been developed to combat these TTR amyloid diseases. The pharmacologic stabilization of the native TTR tetramer using compounds such as the FDA-approved, small molecule drug tafamidis prevents the rate-limiting dissociation of the native TTR tetramer and subsequent aggregation, limiting amyloid toxicity in distal tissues [6,7,8]. Further, genetic reductions in TTR expression in the liver using siRNA or antisense oligonucleotides (ASOs) have also been developed to intervene in TTR-related amyloid disorders [9,10,11,12]. Although these approaches are effective at reducing TTR toxicity in many patients, the positive outcomes of these different approaches are not ubiquitous, necessitating the development of alternative, complementary approaches to ameliorate this disease when alternate approaches fail.

An attractive strategy to mitigate pathologic TTR aggregation is through adaptive remodeling of the endoplasmic reticulum (ER) to reduce the secretion and subsequent aggregation of destabilized TTR variants [13,14,15]. The ER is the first organelle of the secretory pathway responsible for the folding and trafficking of secretory proteins including TTR. In the ER, secretory proteins engage ER-localized protein homeostasis (or proteostasis) factors including chaperones, folding enzymes, and degradation factors that facilitate their partitioning between biological pathways involved in protein folding, trafficking, and degradation—a process referred to as ER quality control [16,17,18]. Quality control functions to facilitate the trafficking of proteins in their native, three-dimensional conformation to downstream secretory environments, while preventing the trafficking of non-native, potentially toxic conformations. Instead, these non-native proteins are directed towards degradation in the ER through mechanisms including ER-associated degradation and ER-phagy [16,17,18].

ER quality control is regulated in response to pathologic ER insults (i.e., ER stress) through the activity of the unfolded protein response (UPR). The UPR comprises three integrated signaling arms activated downstream of the ER stress sensors IRE1, PERK, and ATF6 [18,19,20]. All three arms of the UPR have been implicated in ER quality control regulation. Transcriptional remodeling of ER protein folding, trafficking, and degradation pathways induced by the UPR-associated transcription factors XBP1s (activated downstream of IRE1) and ATF6f (a cleaved product of ATF6) adapt the relative capacities of ER quality control pathways to maintain secretory integrity in response to genetic, environmental, or aging-related stress [15]. Alternatively, PERK signaling influences ER quality control through multiple mechanisms including reductions in ER protein folding load downstream of PERK-induced, eIF2a phosphorylation-dependent translational attenuation [15]. This ability to adapt ER quality control through PERK, IRE1/XBP1s, and/or ATF6 signaling provides a critical mechanism to selectively enhance secretory proteome integrity and protect downstream secretory environments in response to ER stress.

ER stress and UPR signaling have an important role in regulating ER quality control and extracellular aggregation of TTR. ER stress increases secretion of TTR in non-native conformations that accumulates as aggregates in the extracellular space [21,22]. This secretion of non-native TTR is exacerbated by inhibition of the PERK signaling arm of UPR, demonstrating an important role for PERK in regulating TTR secretory proteostasis during ER stress [21]. Age-associated imbalances in UPR signaling in the liver have also been suggested to contribute to the pathologic aggregation and distal deposition of TTR in iPSC and mouse models of disease [23,24]. In contrast, ER remodeling induced by the stress-independent activation of ATF6, but not XBP1s, preferentially reduces the secretion and subsequent aggregation of destabilized aggregation-prone variants of TTR [25,26]. This suggests that enhancing ATF6 activation offers a unique opportunity to mitigate the extracellular TTR aggregation implicated in disease. However, the molecular basis for ATF6-dependent reductions in amyloidogenic TTR secretion and aggregation is poorly understood.

We sought to define how the ATF6-dependent remodeling of ER quality control pathways influences the secretion and subsequent aggregation of amyloidogenic TTR. We show that ATF6 activation does not reduce extracellular TTR aggregation by altering the conformation of TTR secreted from mammalian cells. Instead, ATF6 promotes the ER retention of destabilized TTR through a mechanism involving increased interactions with ATF6-regulated ER chaperones including the ER HSP70 BiP and protein disulfide isomerase PDIA4. Further, we show that the functional interaction between TTR and PDIA4 is not dependent on the single cysteine residue in the TTR sequence or PDIA4 redox activity. This indicates that PDIA4 retains TTR through a redox-independent mechanism. Collectively, these results establish a molecular mechanism to explain the ATF6-dependent reduction in amyloidogenic TTR secretion and subsequent aggregation.

## 2. Materials and Methods

### 2.1. Plasmids, Antibodies, and Reagents

The ^FT^TTR^A25T^ and ^FT^TTR^WT^ plasmids were prepared in the pcDNAI vector as previously reported [26]. The PDIA4 plasmids were prepared in the pCMV vector originally purchased from VectorBuilder, Chicago, IL, USA (Vector ID: VB210118-1178mdb) and mutated using the Takara Infusion kit (Takara Bio, San Jose, CA, USA; CAT# 639643). The BiP plasmid was prepared in the pcDNAI vector as previously reported [26]. Antibodies were purchased from commercial sources: anti-PDIA4 (Proteintech, Rosemont, IL, USA; CAT# 14712-1-AP), anti-KDEL [10C3] (Enzo Life Sciences, Farmingdale, NY, USA; CAT #ADI-SPA827-F), anti-BiP (C50B12) (Cell-Signaling. Danvers, MA, USA; CAT# 3177S), anti-Flag M2 (Sigma Aldrich, St. Louis, MO, USA; CAT #F1804), and anti-α Tubulin (Sigma-Aldrich CAT# T6074). Secondary antibodies for immunoblotting including IRDye Goat anti-mouse 800CW (CAT# 926-32210) and IRDye Goat anti-rabbit 680CW (CAT# 926-68071) were purchased from LI-COR, Lincoln, NE, USA. Anti-FLAG immunopurifications were performed using anti-Flag M1 Agarose Affinity Gel (Sigma Aldrich; A4596). Tafamidis-sulfonate (Taf-S) and compound 1 were generously gifted by Jeffery Kelly at TSRI. DMEM (CAT# 15-017-CM) was purchased from CORNING, Corning, NY, USA. Penicillin/streptomycin (CAT# 15140122), glutamine (CAT# 25030081), and 0.25% Trypsin/EDTA (Cat# 25200056) were purchased from Invitrogen, Waltham, MA, USA. Fetal Bovine Serum was purchased from HyClone, Logan, UT, USA (CAT# SH30396.03; Lot#AF29519994).

### 2.2. Cell Culture, Plasmids, Transfections, and Lysates

HEK293T, HEK293^DAX^, and HepG2 cells were cultured in DMEM-supplemented 10% FBS, penicillin/streptomycin, and glutamine. The cells were transfected with PDIA4 (and mutants), BiP, ^FT^TTR^A25T^, or ^FT^TTR^WT^ plasmids using calcium phosphate transfection, as previously reported [26]. The cells were incubated in transfection media overnight, and the media were replaced with fresh supplemented complete media the next morning. The cells were lysed with RIPA (50 mM Tris pH7.5, 150 mM NaCl, 1% Triton X-100, 0.5% sodium deoxycholate, 10 mM CaCl_2_). The cell lysates were normalized to overall protein concentration in each sample using a Bradford assay.

### 2.3. [^35^S] Metabolic Labeling

Metabolic labeling assays were performed as previously described [25]. Briefly, HEK293T cells transfected with ^FT^TTR^A25T^ and were labeled for 30 min with EasyTag Express [^35^S] Protein Labeling Mix (0.1 mCi/mL; Perkin Elmer CAT# NEG772014MC) in DMEM lacking both Cys and Met (Gibco, Waltham, MA, USA; CAT# 21013024) supplemented with 10% dialyzed FBS, penicillin/streptomycin, and glutamine. Cells were then washed and incubated in complete media. At the indicated time, media and lysates were collected. ^FT^TTR^A25T^ was then immunopurified from the lysate and media using anti-Flag M1 Agarose Affinity Gel (Millipore, Burlington, MA, USA; CAT# A4596) and incubated overnight. We then washed the beads and eluted ^FT^TTR^A25T^ by boiling the beads in 3X Laemmli buffer including 100 mM DTT. Proteins were separated on a 12% SDS-PAGE gel, dried, exposed to phosphorimager plates (GE Healthcare, Chicago, IL, USA), and imaged with a Typhoon imager (Amersham PLC, Amersham, Buckinghamshire, United Kingdom). Band intensities were quantified by densitometry using ImageJ. The fraction secreted was calculated using the equation: fraction secreted = [^35^S] − TTR signal in media at t/([^35^S] − TTR signal in media at t = 0 h + [^35^S] − TTR signal in lysate at t = 0 h). The fraction remaining was calculated using the equation: fraction remaining = ([^35^S] − TTR signal in media at t + [^35^S] − TTR signal in lysate at t)/([^35^S] − TTR signal in media at t = 0 h + [^35^S] − TTR signal in lysate at t = 0 h). The fraction lysate was calculated using the equation: fraction lysate = [^35^S] − TTR signal in lysate at t/([^35^S] − TTR signal in media at t = 0 h + [^35^S] − TTR signal in lysate at t = 0 h).

### 2.4. Media Conditioning, SDS-PAGE, CN-PAGE, and Immunoblotting

HEK293T, HEK293^DAX^, and HepG2 cells transfected with PDIA4 (and mutants), BiP, ^FT^TTR^A25T^, or ^FT^TTR^WT^ were incubated overnight. The following morning, the media were replaced, and cells were incubated for 1 h. Cells were then replated into a poly-D-lysine coated 6-well plate. After 8 h, cells were treated Taf-S for 16 h and conditioned in 1 mL of media. Conditioned media were then collected and cleared from large debris by centrifugation at 1000× *g* for 10 min. For SDS-PAGE, samples were boiled in Lammeli Buffer containing 100 mM DTT for 5 min and resolved on a 12% acrylamide gel. The gels were transferred to 0.2 µm nitrocellulose membranes at 100 V for 70 min. For CN-PAGE, the samples were added to 5X Clear Native Page Buffer (final concentration: 0.1% Ponceau Red, 10% glycerol, 50 mM 6-aminohexanoic acid, 10 mM Bis-Tris pH 7.0) and resolved on a Novex NativePage 4–16% Bis-Tris Protein Gel (Invitrogen, Waltham, MA, USA; CAT# BN1004BOX) run at 150 V for 3 h. The protein was transferred at 100 V for 75 min to a 0.2 µm nitrocellulose membrane. The membranes were incubated with the indicated primary antibody overnight and then incubated with the appropriate LI-COR secondary antibody. The protein bands where then visualized and quantified using the LI-COR Odyssey Infrared Imaging System Image Studio software (LI-COR, Lincoln, NE, USA; Version 5.2).

### 2.5. Quantification of ^FT^TTR Tetramers in Conditioned Media

^FT^TTR tetramers in conditioned media were quantified as previously described [22]. Briefly, conditioned media were prepared on HEK293T cells expressing the indicated ^FT^TTR variant treated with Taf-S overnight. Conditioned media were then incubated with 5 µM compound 1 overnight. Media samples (60 µL) were separated over a Waters Protein-Pak Hi Res Q, 5 µm 4.6 × 100 mm anion exchange column (Waters, Milford, MA, USA; CAT# 186004931) in 25 mm Tris pH 8.0, 1 mM EDTA with a linear 1 M NaCl gradient using a ACQUITY UPLC H-Class Bio System (Waters, Milford, MA, USA). The fluorescence of TTR tetramers conjugated to compound 1 was observed by excitation at 328 nm and emission at 430 nm. Peaks were integrated and data were collected using Empower 3 software (Waters, Milford, MA, USA; Version 3471) according to the manufacture’s protocol.

### 2.6. Co-Immunoprecipitation of ^FT^TTR^A25T^

HEK293T and HEK293^DAX^ cells transfected with PDIA4 (and mutants) and ^FT^TTR^A25T^ were incubated overnight. The following morning, the media were replaced, and the cells were incubated for another 24 h. The following morning, the cells were collected and crosslinked with 500 µM dithiobis (succinimidyl propionate) (DSP) (Thermo Scientific, Waltham, MA, USA; CAT# 22585) for 30 min at RT rocking. The reaction was quenched with 100 mM Tris Base pH 7.5 (Fisher BioReagents, Pittsburgh, PA, USA; Cat# BP152) for 15 min at RT rocking. The cells were then lysed with RIPA as previously described. The lysates were placed on Sepharose 4B beads (Sigma-Aldrich, St. Louis, MO, USA; CAT# 4B200) to clear for 30 min at 4 °C rocking. Then, the cleared lysate was placed on M1-FLAG Sepharose beads overnight at 4 °C rocking. The following morning, the beads were washed 4 times with RIPA at 4 °C for 10 min each. The beads were then eluted with 3X LB 100 mM DTT for 10 min at 95 °C, and the supernatant was subsequently loaded on a 12% acrylamide gel for analysis. All proteins pulled down were normalized to the amount of ^FT^TTR in each condition.

### 2.7. SILAC-MuDPIT Proteomics

Light and heavy HEK293^DAX^ cells were prepared as conducted previously [26]. The cells were grown in SILAC media, with light cells supplemented with light lysine and arginine and heavy cells supplemented with ^13^C6-lysine and ^13^C6-^15^N4-arginine. Each cell line was seeded at 15% confluency in fifteen 10 cm plates and transfected the next day with ^FT^TTR^A25T^ (all heavy plates and 12 plates of light) or No Bait (3 light plates of TTR^A25T^). The media were changed the following day. Heavy cells were not further treated. Three plates each of light cells were treated with vehicle (0.01% DMSO), TMP (10 µM), dox (1 µg/mL) + 0.01% DMSO, or TMP and dox, for 16 h. The cells were harvested, and the heavy lysates were pooled and quantified and then added to each light lysate at a 1:1 total protein ratio and brought to 2 mL total volume with RIPA. Immunoprecipitation followed as stated in the above methods. Proteins were eluted in 50 µL cold 8 M Urea in 50 mM Tris pH 8 at 4 °C overnight. Eluates were chloroform methanol precipitated and resuspended in freshly prepared 8 M urea in 50 mM Tris pH 7.5. TCEP was added to 10 mM, followed by a 30 min incubation at room temperature. Iodoacetamide was added to a final concentration of 12 mM, followed by a 30 min incubation at room temperature in the dark. The samples were digested overnight with 0.5 µg trypsin (Promega) at 27 °C and 600 rpm. The samples were spun at 16,000× *g* for 30 min prior to loading onto an equilibrated MuDPIT column. The loaded MuDPIT columns were washed well with buffer A (5% ACN/95% water/0.1% formic acid). At this point, only mass-spectrometry-grade solvents were used. MuDPIT columns were prepared from 250 µM fused silica capillaries (Agilent) by polymerizing a Kasil 1624 frit and cutting to leave about 2 mm frit. The column was filled with 2.5 cm of reversed-phase 5 µm C18 resin (Phenomenex, Torrance, CA, USA; CAT# 04A-4299) followed by 2.5 cm of strong cation exchange resin, and another 2.5 cm of reversed-phase 5 µm C18 resin. Analytical columns were pulled from 100 µm fused silica capillaries using a P-2000 Sutter Instruments tip puller and filled with about 15 cm C18 resin. The columns were washed with methanol and thoroughly equilibrated with buffer A. Online LC was performed with an HPLC pump 1200 (Agilent Technologies, La Jolla, CA, USA) using five-step MuDPIT. The initial run consisted of increasing acetonitrile, which moved the peptides from the C18 resin to the strong cation exchange resin. This was followed by 4 salt bump runs, injecting 10 µL of 25%, 50%, 75%, and 100% 10 mM ammonium acetate, respectively, with a gradient from buffer A to buffer B (100% acetonitrile + 0.1% formic acid). MS was performed on a LTQ Orbitrap Velos Pro ion trap mass spectrometer (Thermo Scientific, Waltham, MA, USA) with spray voltage set to 2.50 kV, running DDA with each full Orbitrap MS Survey scan (30,000 resolving power) followed by 10 MS/MS scans in the ion trap. Dynamic exclusion was enabled with a repeat count of 1, a repeat duration of 30 s, an exclusion duration of 120 s, and an exclusion list size of 500. Peptides were fragmented by CID with a normalized collision energy of 35% and a 2 Da isolation window. Spectra were extracted from the raw files using Raw Extractor and searched with ProLucid in the IP2 engine with fixed modification at cysteine (57.02146 Da) and variable modifications at lysine and arginine. Searches were performed against a UniProt protein dataset (URL: https://www.uniprot.org/proteomes/UP000005640) (accessed on 28 March 2012, with Flag-TTR-A25T appended, curated, single isoform) with appended reverse decoy sequences. Protein hits required at least two peptides and were filtered in DTASelect to ensure ≤1% FDR at the peptide level. SILAC intensities were integrated in Census. All data analysis above was performed using the IP2-Integrated Proteomics Pipeline developed at The Scripps Research Institute (URL: IP2.scripps.edu).

## 3. Results

### 3.1. Stress-Independent ATF6 Activation Does Not Influence the Conformation of Secreted ^FT^TTR^A25T^

ER stress increases the population of destabilized TTR variants secreted in non-native conformations, accelerating extracellular accumulation of soluble TTR aggregates [21,22]. This suggests that alterations in ER proteostasis can influence the extracellular aggregation of TTR by altering the conformation of TTR secreted from mammalian cells. Stress-independent ATF6 activation reduces the secretion and subsequent accumulation of TTR aggregates in cell culture models [25,26]. However, the impact of ATF6 activation on the conformation of TTR secreted from mammalian cells has not been previously determined. To address this, we monitored the population of tetrameric, aggregate, and total flag-tagged TTR^A25T^ (^FT^TTR^A25T^) in conditioned media prepared on HEK293^DAX^ cells, using established assays (Appendix A). ^FT^TTR^A25T^ is a destabilized TTR variant whose secretion and extracellular aggregation is highly sensitive to alterations in ER quality control [21,22,27,28]. HEK293^DAX^ cells are a stable cell line expressing both doxycycline (dox)-inducible XBP1s and trimethoprim (TMP)-regulated DHFR.ATF6, allowing ligand-dependent activation of XBP1s or ATF6 signaling independent of ER stress through administration of dox or TMP [26]. Total TTR in conditioned media was quantified by SDS-PAGE/immunoblotting and aggregate TTR was monitored by Clear-Native (CN) Page/immunoblotting (Appendix A) [21,22]. Tetrameric TTR was monitored using the covalent TTR ligand, compound **1** (Appendix A), which fluoresces upon binding to TTR tetramers [21,22,29]. The relative population of TTR tetramers was then quantified by monitoring compound **1**-native TTR conjugate fluorescence during anion exchange ultraperformance liquid chromatography (UPLC) [21,22]. To prevent the dissociation and subsequent aggregation of secreted TTR tetramers in these experiments, we conditioned media in the presence of the cell impermeable, non-covalent TTR ligand tafamidis-sulfonate (Taf-S), which binds to TTR tetramers immediately upon secretion, inhibiting both tetramer dissociation and subsequent aggregation [21,22]. Taf-S can be displaced by compound **1** (because it binds and reacts with TTR) to enable later native tetramer quantification [21,22].

Stress-independent ATF6 activation reduced total ^FT^TTR^A25T^ in media conditioned on HEK293^DAX^ cells by 30% (Figure 1A,B). However, XBP1s activation did not reduce ^FT^TTR^A25T^ secretion to a significant extent (Appendix A). This is consistent with previous results [25,26]. ATF6 activation similarly reduced tetrameric and aggregate TTR in conditioned media by 30% (Figure 1A–D and Appendix A). Normalizing the amount of TTR tetramers and aggregates in conditioned media to total TTR showed no discernable difference upon ATF6 activation (Figure 1E,F). This indicates that ATF6 activation did not significantly impact the conformation of secreted TTR. In contrast, thapsigargin (Tg)-induced ER stress reduced normalized TTR tetramers in conditioned media and increased the population of secreted TTR aggregates (Figure 1A–F). This is consistent with previous results showing that Tg increased the secretion of TTR in non-tetrameric conformations that accumulate as aggregates in the extracellular space [21,22]. These results show that ATF6 activation does not influence the extracellular aggregation of destabilized ^FT^TTR^A25T^ by impacting the conformation of the secreted protein. Instead, ATF6 activation decreases TTR aggregates in conditioned media by reducing the total amount of secreted protein.

### 3.2. ATF6 Activation Increases Interaction between ^FT^TTR^A25T^ and ER-Localized Proteostasis Factors

ATF6 or XBP1s transcriptional activation induces the expression of distinct, but overlapping, sets of ER proteostasis factors that differentially influence ER quality control of destabilized proteins **[26]**. To further define how ATF6 activation reduces extracellular TTR secretion and subsequent aggregation, we employed a SILAC-based quantitative proteomic approach to identify ER proteostasis factors that show increased interactions with ^FT^TTR^A25T^ following stress-independent activation of ATF6 or XBP1s in HEK293^DAX^ cells (Appendix A). Initially, we defined ER proteostasis factors that engage ^FT^TTR^A25T^ by comparing proteins recovered in FLAG immunopurifications (IPs) from HEK293^DAX^ cells that were either mock transfected or transfected with ^FT^TTR^A25T^. 

We identified 65 ‘true’ interactors that show a 2-fold increase in recovery from ^FT^TTR^A25T^ relative to mock IPs (Figure 2A, Appendix A). The majority of these interactors are localized to the ER and have known ER proteostasis functions. Next, we compared the recovery of these ‘true’ interactors in ^FT^TTR^A25T^ IPs from HEK293^DAX^ cells following stress-independent ATF6 or XBP1s activation to that observed in vehicle-treated cells. The quantified amount of true interactors in these ^FT^TTR^A25T^ IPs was normalized to the quantified amount of ^FT^TTR^A25T^, allowing us to identify proteins that showed altered interactions with ^FT^TTR^A25T^ independent of changes to intracellular TTR induced by these conditions. The stress-independent activation of ATF6, but not XBP1s, increased interactions with multiple ER proteostasis factors, including the ER HSP70 BiP and the protein disulfide isomerase PDIA4 (Figure 2B, Appendix A). We confirmed ATF6-dependent increases in interactions between ^FT^TTR^A25T^ and BiP or PDIA4 by IP/immunoblotting (Figure 2C,D). Interestingly, proteins that show increased interaction with ^FT^TTR^A25T^ are known transcriptional targets of the ATF6 arm of the UPR [26]. This suggested that ATF6 activation increases interactions between ^FT^TTR^A25T^ and these ER proteostasis factors through their increased expression. Consistent with this, changes in interactions between ^FT^TTR^A25T^ and ER proteostasis factors induced by TMP-dependent DHFR.ATF6 activation in HEK293^DAX^ cells correlated with the ATF6-dependent increase in their expression, measured by RNAseq (Figure 2E). This indicates that the observed changes in interactions between ^FT^TTR^A25T^ and ER proteostasis factors such as BiP and PDIA4 can be largely attributed to increases in their expression following ATF6 activation.

### 3.3. Overexpression of BiP or PDIA4 Reduces Secretion and Extracellular Aggregation of Destabilized ^FT^TTR^A25T^

We next sought to define how increases in BiP or PDIA4 protein impact the secretion and extracellular aggregation of destabilized ^FT^TTR^A25T^. The depletion of *BiP* or *PDIA4* causes ER stress and UPR activation [30,31]. Instead, we overexpressed these two ER quality control factors in HEK293T cells and monitored total and aggregate ^FT^TTR^A25T^ in conditioned media. The overexpression of BiP or PDIA4 reduced the total secreted ^FT^TTR^A25T^ by 70% and 30%, respectively (Figure 3A,B). Similar reductions were observed for TTR aggregates in conditioned media (Figure 3A,C). Normalizing the amount of observed aggregate to the total amount of TTR in conditioned media for each condition shows that neither the overexpression of BiP nor PDIA4 influenced the conformation of secreted TTR (Appendix A), mirroring the results observed following ATF6 activation (Figure 1E). PDIA4 overexpression did not significantly influence the secretion of wild-type ^FT^TTR, indicating that overexpressing this protein preferentially impacts secretion of destabilized, amyloidogenic TTRs (Appendix A). This is consistent with previous results showing that ATF6 activation preferentially reduces the secretion of destabilized, aggregation-prone TTR variants [25].

We next determined whether PDIA4 or BiP overexpression could, on its own, explain the ATF6-dependent reduction in destabilized ^FT^TTR^A25T^ secretion. We overexpressed these two ER proteostasis factors in HEK293^DAX^ cells and monitored total ^FT^TTR^A25T^ secretion in conditioned media prepared on cells treated with TMP (activates ATF6) and/or doxycycline (activates XBP1s). Interestingly, the activation of ATF6, but not XBP1s, further reduced total secreted ^FT^TTR^A25T^ in media prepared on cells expressing PDIA4 or BiP (Appendix A). This suggests that ATF6 activation reduces destabilized TTR secretion in HEK293^DAX^ cells through a mechanism involving the increased expression of multiple ER proteostasis factors including both PDIA4 and BiP.

To evaluate whether PDIA4 or BiP overexpression similarly reduced amyloidogenic TTR secretion in an amyloid disease relevant cell line, we monitored total and aggregate ^FT^TTR^A25T^ in conditioned media prepared on liver-derived HepG2 cells. Surprisingly, BiP overexpression did not significantly decrease ^FT^TTR^A25T^ secretion or extracellular aggregation in HepG2 cells (Figure 3D–F). However, PDIA4 overexpression significantly reduced total and aggregate ^FT^TTR^A25T^ in media prepared on these cells (Figure 3D–F). This reduction in ^FT^TTR^A25T^ secretion from HepG2 cells corresponds with increases in lysate ^FT^TTR^A25T^, suggesting that PDIA4 may reduce amyloidogenic TTR in the ER through increased ER retention (Figure 3D and Appendix A). These results indicate that increases in PDIA4 can reduce secretion and subsequent aggregation of destabilized, amyloidogenic TTR in multiple cell models, including liver-derived HepG2 cells.

### 3.4. PDIA4 Overexpression Increases ER Retention of ^FT^TTR^A25T^

We further defined the impact of PDIA4 and BiP overexpression on destabilized ^FT^TTR^A25T^ secretion using a [^35^S] metabolic labeling pulse-chase assay in HEK293T cells (Figure 4A). The overexpression of BiP reduced secreted [^35^S]-labeled ^FT^TTR^A25T^ by 50% 4 h after labeling (Figure 4A,B). This corresponded with both a modest increase in ^FT^TTR^A25T^ degradation (evident by reductions in total ^FT^TTR^A25T^ in both media and lysate at 4 h in BiP-overexpressing cells relative to mock-transfected cells; Figure 4C) and an increase in ^FT^TTR^A25T^ ER retention (evident by higher amounts of ^FT^TTR^A25T^ in lysates at 4 h in BiP overexpressing cells relative to mock transfected cells; Figure 4D). The increase in ^FT^TTR^A25T^ ER retention afforded by BiP overexpression is consistent with previous results showing that BiP retains destabilized TTRs within the ER lumen [32]. Interestingly, PDIA4 overexpression similarly reduced ^FT^TTR^A25T^ secretion by 50% (Figure 4A,B). This reduced secretion also correlated with an increased ER retention of ^FT^TTR^A25T^ (Figure 4D). This suggests that the reduction in destabilized TTR secretion afforded by PDIA4 overexpression is mediated through a mechanism involving increased ER retention.

### 3.5. PDIA4-Dependent Reductions in ^FT^TTR^A25T^ Secretion Is Independent of Cys10

PDIA4 is a protein disulfide isomerase that primarily functions to facilitate the proper assembly of native disulfide bonds in secretory proteins as they are processed within the ER [33]. However, TTR lacks a disulfide bond in the native tetramer and only contains a single cysteine residue at position 10 (Cys10) of the mature protein. PDIA4 can assemble into a complex comprising multiple other ER proteostasis factors including BiP [34]. This suggests that PDIA4 could bind to destabilized TTR either directly or in a complex including other proteins. To probe the potential for PDIA4 and BiP to engage destabilized TTR as part of a complex, we compared the relative recovery of PDIA4 and BiP in ^FT^TTR^A25T^ IPs prepared from HEK293T cells overexpressing PDIA4. We observed a ~100-fold increase in PDIA4 in ^FT^TTR^A25T^ IPs upon the overexpression of PDIA4 (Appendix A). In contrast, we observed a more modest ~5-fold increase in BiP from these IPs. While this indicates that PDIA4 overexpression can modestly impact interactions between TTR and other ER proteostasis factors, the differential recovery of PDIA4 and BiP in ^FT^TTR^A25T^ IPs under these conditions suggest that their interactions with destabilized TTR are largely independent. Although these results do not exclude the possibility that PDIA4 engages destabilized TTR in complex with other ER proteostasis factors, they are consistent with a model whereby PDIA4 can bind destabilized TTR directly in the ER.

Next, to determine whether Cys10 is required for PDIA4-dependent reductions in amyloidogenic TTR secretion, we monitored total and aggregate concentrations of a TTR construct containing both the destabilizing A25T mutation and a C10A mutation (^FT^TTR^C10A,A25T^) in media prepared on HEK293T cells overexpressing PDIA4 or BiP. We found that the overexpression of either PDIA4 or BiP reduced total ^FT^TTR^C10A,A25T^ in conditioned media by 30% and 70%, respectively (Figure 5A,B). Identical reductions were observed in aggregates of ^FT^TTR^C10A,A25T^ (Figure 5A,C). These changes mirror the reductions in ^FT^TTR^A25T^ afforded by the overexpression of these ER quality control factors (Figure 3A–C). This indicates that PDIA4-dependent reductions in amyloidogenic TTR secretion and aggregation are independent of Cys10.

### 3.6. Overexpression of Redox-Deficient PDIA4 Variants Decrease ^FT^TTR^A25T^ Secretion and Aggregation

Other ER PDIs, including PDIA1, have been reported to influence ER quality control for secretory proteins through redox-independent chaperoning activities [35,36]. Thus, to determine whether PDIA4 was influencing ^FT^TTR^A25T^ secretion through a redox-independent mechanism, we monitored total and aggregate TTR in media prepared on HEK293T cells expressing redox-deficient PDIA4 variants. PDIA4 is a multi-domain protein containing three domains: the catalytically active *a*-domain containing two redox active sites (*a*1 and *a*2) comprising the sequence CGHC, the catalytically inactive *b*-domain, and a substrate binding *a’*-domain that contains a third CGHC redox active site (Figure 6A) [33]. We mutated the two active site Cys residues to Ser (SGHS) at each of these sites both individually and in combination (Figure 6A). We then overexpressed these variants in HEK293T cells and monitored their impact on total and aggregate ^FT^TTR^A25T^ in conditioned media. All three PDIA4 variants expressing mutant redox active sites at the *a*1-, *a*2-, or *a’*-domains showed the same 4-fold increases in expression observed for wild-type PDIA4 when overexpressed in HEK293T cells (Figure 6B and Appendix A). These protein level changes are similar to the 3-fold increase in PDIA4 observed upon TMP-dependent DHFR-ATF6 activation in HEK293^DAX^ cells [26]. These variants also showed similar recovery in ^FT^TTR^A25T^ IPs from these cells (Appendix A). However, the triple mutant (*a*1*a*2*a’*^Mut^) showed lower levels of expression and reduced recovery in ^FT^TTR^A25T^ IPs relative to the other variants, potentially reflecting the reduced stability of this PDIA4 mutant. Regardless, all four PDIA4 variants decreased the total and aggregate ^FT^TTR^A25T^ in conditioned media prepared on HEK293T cells (Figure 6C,D). We also observed similar increases in lysate ^FT^TTR^A25T^ in cells overexpressing these PDIA4 mutants (Appendix A), suggesting these mutants induce the same increased ER retention observed upon wild-type PDIA4 overexpression. Collectively, these results indicate that the reductions in ^FT^TTR^A25T^ secretion afforded by PDIA4 overexpression are independent of PDIA4 redox activity.

## 4. Discussion

Stress-independent activation of ATF6 has emerged as a promising approach to reduce secretion and subsequent aggregation of amyloidogenic proteins such as TTR [13,14,15]. However, the molecular basis for ATF6-dependent reductions in amyloidogenic TTR secretion was previously poorly understood. Here, we demonstrate that ATF6 activation decreases secretion of a destabilized, amyloidogenic TTR variant through a mechanism involving increased interactions with ATF6-regulated ER chaperones and folding factors including BiP and PDIA4 that retain the aggregation prone TTR within the ER. Mimicking the ATF6-dependent increases in BiP and PDIA4 expression through overexpression reduces the secretion and extracellular aggregation of amyloidogenic TTR through increased ER retention. Intriguingly, we demonstrate that the PDIA4-dependent reduction in amyloidogenic TTR secretion is independent of both the single Cys residue within TTR and PDIA4 redox activity, indicating that this decrease is attributed to a redox-independent mechanism. These results reveal new insights into the importance of ER quality control pathways in regulating secretion and extracellular aggregation of amyloidogenic TTRs. This further highlights the potential for enhancing the activity of key ER quality control factors including BiP or PDIA4 through mechanisms such as ATF6 activation to ameliorate pathologic extracellular aggregation of TTR implicated in amyloid disease pathogenesis.

Our results demonstrate that ATF6 reduces the secretion of amyloidogenic TTR through a mechanism involving increased expression of multiple ER folding factors that retain the destabilized TTR within the ER. This is a similar mechanism to that shown for amyloidogenic immunoglobulin light chains (LCs), where increased interactions with ATF6-regulated ER chaperones including BiP preferentially retain destabilized LCs within the ER [37]. This indicates that the ATF6-dependent upregulation of BiP is a critical mechanism to enhance ER quality control for structurally distinct destabilized, amyloidogenic proteins. The importance of BiP for TTR folding, aggregation, and secretion has been previously demonstrated. BiP preferentially binds destabilized, aggregation-prone TTR variants [38]. Further, BiP overexpression negatively regulates the degradation of highly destabilized TTR variants in the ER through increased ER retention [32,39]. Our results support this role of BiP-dependent retention for regulating the secretion of destabilized, aggregation-prone TTR variants. However, the inability for BiP overexpression to influence ^FT^TTR^A25T^ secretion in liver-derived HepG2 cells indicates that the effect of BiP is likely dependent on the basal ER quality control activity of a given cell.

In contrast, PDIA4 overexpression reduces amyloidogenic TTR secretion in both HEK293T and HepG2 cells, suggesting an important role for this ER quality control factor in regulating TTR secretion. However, PDIA4 overexpression did not influence the secretion of stable, wild-type TTR, indicating that PDIA4 preferentially influences secretion of destabilized, amyloidogenic TTR. The importance of PDIA4 on TTR secretion was surprising, considering that TTR does not contain a disulfide in the native tetrameric protein. Instead, this suggested that PDIA4-dependent reductions in TTR secretion were mediated through redox-independent chaperoning activity. Consistent with this, we show that the mutation of the sole TTR Cys residue or removal of the three redox active sites of PDIA4 do not impair PDIA4-dependent reductions in amyloidogenic TTR secretion. A similar redox-independent activity was previously reported for other PDIs including PDIA1 [35,36]; however, we are unaware of any previous reports showing a similar activity for PDIA4. Our results indicate that PDIA4 can also regulate the secretion of proteins such as TTR through this type of redox independent mechanism and that increasing PDIA4 expression through ATF6 activation can exploit this activity to improve ER quality control for destabilized, amyloidogenic proteins.

PDIA4 can assemble into a complex comprising multiple other ER proteostasis factors including BiP [34]. Our results indicate that increased PDIA4 association with ^FT^TTR^A25T^ in cells overexpressing PDIA4 are largely independent of increased interactions with BiP. This suggests that PDIA4 could bind destabilized TTR directly; however, we cannot rule out the possibility that PDIA4 engages destabilized TTR as part of a complex with other ER proteostasis factors. Regardless, we show that increases in PDIA4 reduce amyloidogenic TTR secretion in both HEK293T cells and liver-derived HepG2 cells, while BiP overexpression only reduces destabilized TTR secretion in HEK293T cells. Further, we demonstrate that stress-independent ATF6 activation further reduces amyloidogenic TTR secretion in HEK293T cells overexpressing either PDIA4 or BiP. This likely reflects the cooperation between multiple ATF6-regulated ER proteostasis factors (e.g., BiP and PDIA4) in regulating amyloidogenic TTR secretion and highlights the global impact of activating this transcription factor in the remodeling of ER quality control [18]. Taken together, our results indicate an important role for PDIA4 in adapting TTR ER quality control and suppressing the secretion and subsequent aggregation of destabilized, aggregation-prone TTR variants.

Reducing the secretion of destabilized, amyloidogenic proteins like TTR is an attractive strategy to mitigate pathologic concentration-dependent aggregation and subsequent deposition of aggregates on distal tissues implicated in multiple systemic amyloid diseases [37,40]. This approach is complementary to other strategies to intervene in these diseases such as the pharmacologic kinetic stabilization of native protein conformations exemplified by the binding of tafamidis to native TTR tetramers. Consistent with this, tafamidis-dependent TTR tetramer stabilization and ATF6-dependent reductions in amyloidogenic TTR secretion both decrease the accumulation of TTR aggregates in conditioned media through distinct mechanisms [25]. Our results reveal mechanistic insights into the ATF6-dependent reduction in destabilized TTR secretion that could be further exploited to decrease pathologic TTR aggregation. For example, strategies that enhance PDIA4 chaperoning activity could offer new opportunities to preferentially reduce the secretion and toxic aggregation of TTR in the context of TTR amyloid diseases. Further, our results establish a molecular framework for the further development of pharmacologic ER proteostasis regulators, such as ATF6-activating compounds [41,42,43], that can be tailored to optimize reductions in the secretion of destabilized, amyloidogenic TTR variants that are responsible for disease onset.

## Figures and Tables

**Figure 1 cells-11-01661-f001:**
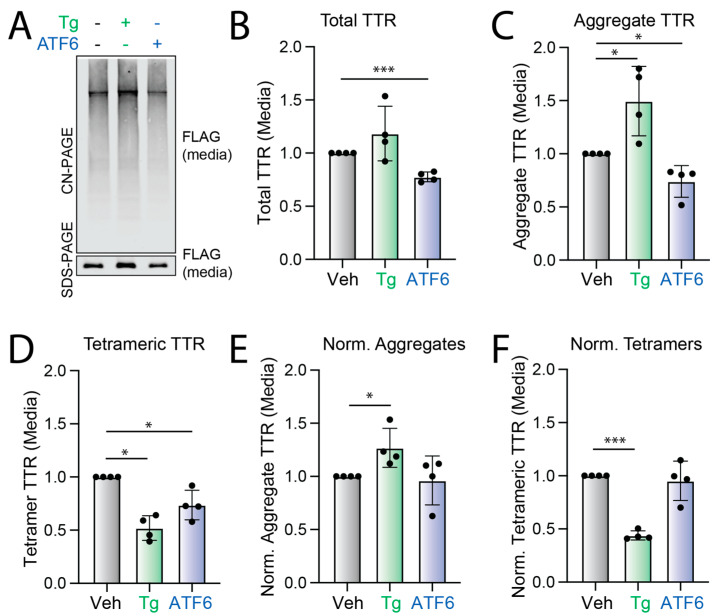
ATF6 activation does not influence the conformation of secreted ^FT^TTR^A25T^. (**A**–**C**) Representative Clear Native (CN)-PAGE and SDS-PAGE immunoblots and quantification of total and aggregate ^FT^TTR^A25T^ in conditioned media prepared on transiently transfected HEK293^DAX^ cells treated for 18 h with thapsigargin (Tg; 500 nM) or trimethoprim (TMP, 10 µM; activates ATF6). Media were conditioned in the presence of Tafamidis-sulfonate (Taf-S; 10 µM). Quantifications in (**B**,**C**) are shown normalized to vehicle (Veh)-treated cells. Error bars show SEM for *n* = 4 replicates. (**D**) Tetrameric ^FT^TTR^A25T^, measured by compound **1** fluorescence, in conditioned media described in panels (**A**–**C**). Data are shown normalized to vehicle (Veh) treated cells. A representative chromatogram is shown in Appendix A. Error bars show SEM for *n* = 4 replicates. (**E**) Normalized TTR aggregates in the conditioned media described in panels (**A**–**C**). Normalized TTR aggregates were calculated using the following formula: normalized TTR aggregates = ([aggregate TTR in a given condition]/[aggregate TTR in media from veh-treated cells])/([total TTR in a given condition]/[total TTR in media from veh-treated cells]). Error bars show SEM for *n* = 4 independent replicates. (**F**) Normalized TTR tetramers in the conditioned media described in panels (**A**–**C**). Normalized TTR tetramers were calculated using the following formula: normalized TTR tetramers = ([tetrameric TTR in a given condition]/[tetrameric TTR in media from veh-treated cells])/([total TTR in a given condition]/[total TTR in media from veh-treated cells]). Error bars show SEM for *n* = 4 independent replicates * *p* < 0.05, *** *p* < 0.001 for an RM one-way ANOVA relative to Veh-treated cells.

**Figure 2 cells-11-01661-f002:**
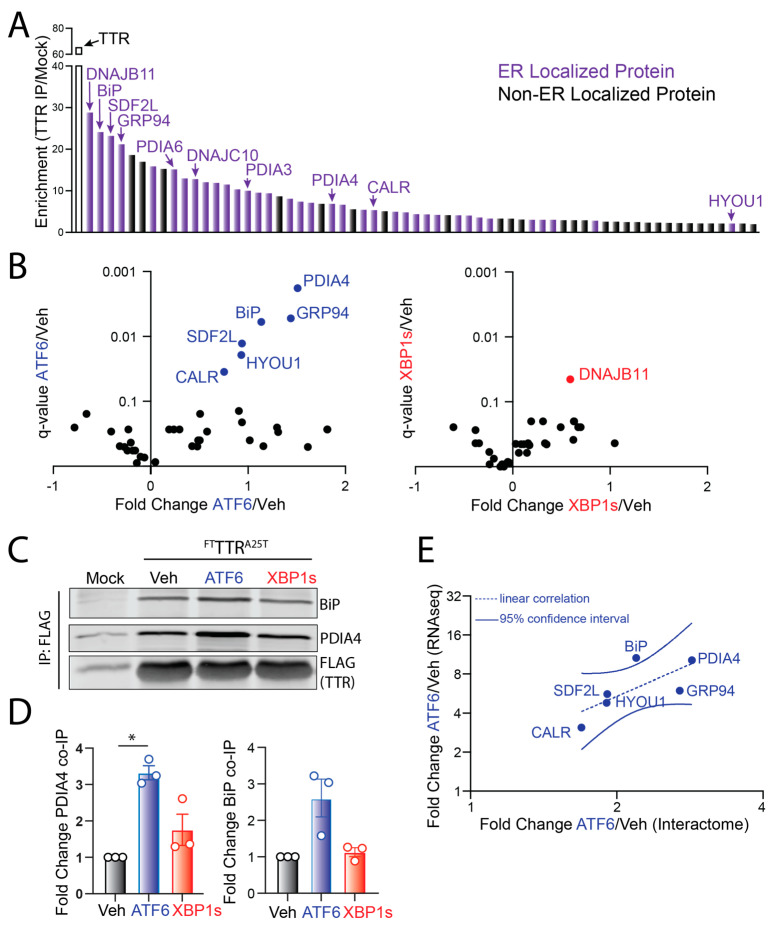
ATF6 activation increases interactions between ^FT^TTR^A25T^ and ER-localized proteostasis factors. (**A**) SILAC-based quantitative proteomics comparing proteins recovered in FLAG immunopurifications (IPs) from lysates prepared from HEK293^DAX^ cells expressing empty vector or ^FT^TTR^A25T^. ER-localized proteins identified are shown in purple. Proteins shown to have a >2-fold enrichment in ^FT^TTR^A25T^ IPs are considered ‘true interactors’. Data are included in Appendix A. (**B**) Plots comparing the log_2_ fold change and *q*–value for ‘true ^FT^TTR^A25T^ interactors’ from lysates prepared from HEK293^DAX^ following treatments with trimethoprim (TMP; activates ATF6, left) or doxycycline (dox; activates XBP1s, right), as compared to vehicle-treated cells. Data are included in Appendix A. (**C**,**D**) Representative immunoblot and quantification of BiP and PDIA4 in FLAG IPs from HEK293^DAX^ cells transiently expressing mock or ^FT^TTR^A25T^ following 18 h treatment with trimethoprim (TMP; activates ATF6) or doxycycline (dox; activates XBP1s). Error bars show SEM for *n* = 3 replicates. (**E**) Comparison of fold change increase in mRNA, measured by RNAseq, induced by TMP-dependent ATF6 activation versus fold change increase in protein induced by TMP-dependent ATF6 activation in HEK293^DAX^ cells for proteins shown to increase association with ^FT^TTR^A25T^ in FLAG IPs from HEK293^DAX^ cells following TMP-dependent ATF6 activation. * *p* < 0.05 for an RM one-way ANOVA relative to Veh-treated cells.

**Figure 3 cells-11-01661-f003:**
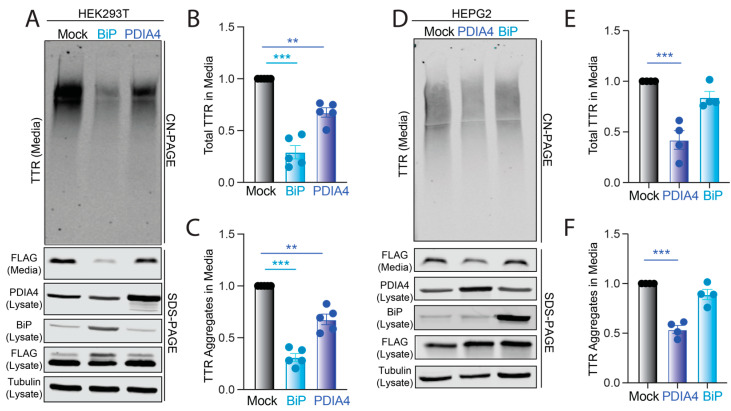
BiP or PDIA4 overexpression reduces total and aggregate ^FT^TTR^A25T^ in conditioned media. (**A**–**C**) Representative CN-PAGE and SDS-PAGE immunoblot and quantification of total and aggregate ^FT^TTR^A25T^ in conditioned media prepared on transiently transfected HEK293T cells overexpressing mock, BiP or PDIA4. Media were conditioned for 18 h. Error bars show SEM for *n* = 5 independent replicates. (**D**–**F**) Representative CN-PAGE and SDS-PAGE immunoblot and quantification of total and aggregate ^FT^TTR^A25T^ in conditioned media prepared on transiently transfected HepG2 cells overexpressing mock, BiP or PDIA4. Media were conditioned for 18 h. Error bars show SEM for *n* = 4 independent replicates. ** *p* < 0.01, *** *p* < 0.005 for an RM one-way ANOVA relative to Mock-transfected cells.

**Figure 4 cells-11-01661-f004:**
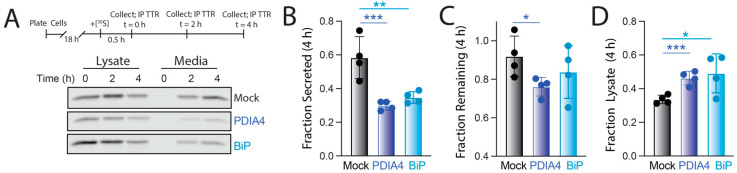
PDIA4 or BiP overexpression increases ER retention of destabilized ^FT^TTR^A25T^. (**A**) Representative autoradiogram of [^35^S]-labeled ^FT^TTR^A25T^ immunopurified from lysates and media prepared on HEK293T cells overexpressing mock, PDIA4, or BiP 0, 2, or 4 h after a 30 min pulse of [^35^S]-Met. The experimental protocol is shown above. (**B**–**C**) Fraction secreted (**B**), Fraction remaining (**C**), and Fraction lysate (**D**) of [^35^S]-^FT^TTR^A25T^ at 4 h under the same conditions described in (**A**). Fraction secreted, remaining, and lysate were calculated as described in Materials and Methods. Error bars show SEM for *n* = 4 independent replicates. * *p* < 0.05, ** *p* < 0.01, *** *p* < 0.005 for an RM one-way ANOVA relative to Mock-transfected cells.

**Figure 5 cells-11-01661-f005:**
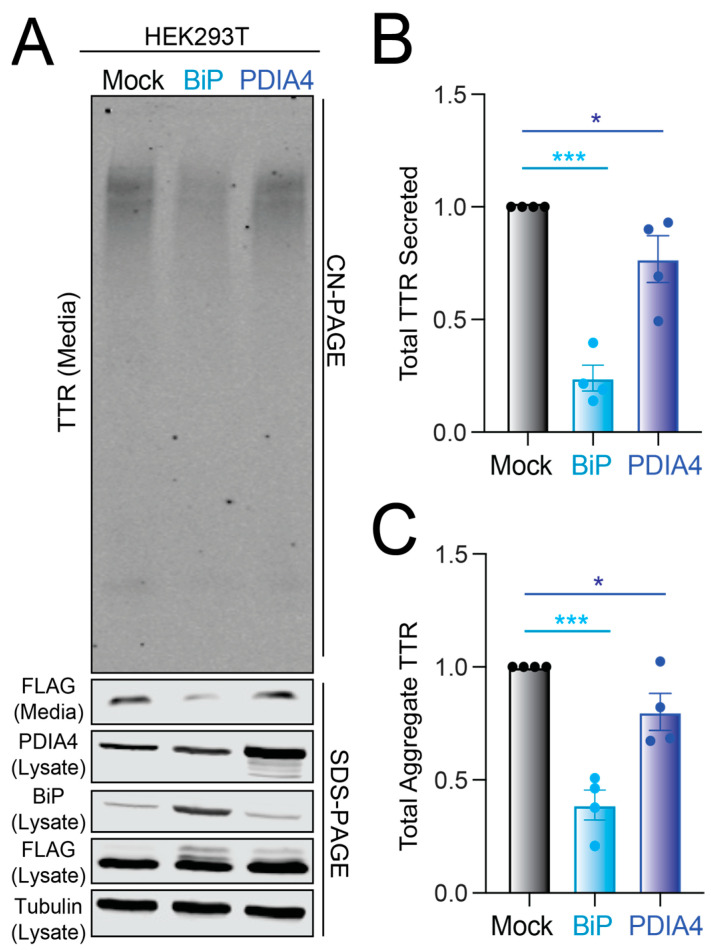
PDIA4 overexpression reduces secretion and aggregation of ^FT^TTR^A25T^ independently of Cys10 of TTR. (**A**–**C**) Representative CN-PAGE and SDS-PAGE immunoblot and quantification of total and aggregate ^FT^TTR^C10A,A25T^ in conditioned media prepared on transiently transfected HEK293T cells overexpressing mock, BiP or PDIA4. Media were conditioned for 18 h. Error bars show SEM for *n* = 4 independent replicates. * *p* < 0.05, *** *p* < 0.005 for an RM one-way ANOVA relative to Mock-transfected cells.

**Figure 6 cells-11-01661-f006:**
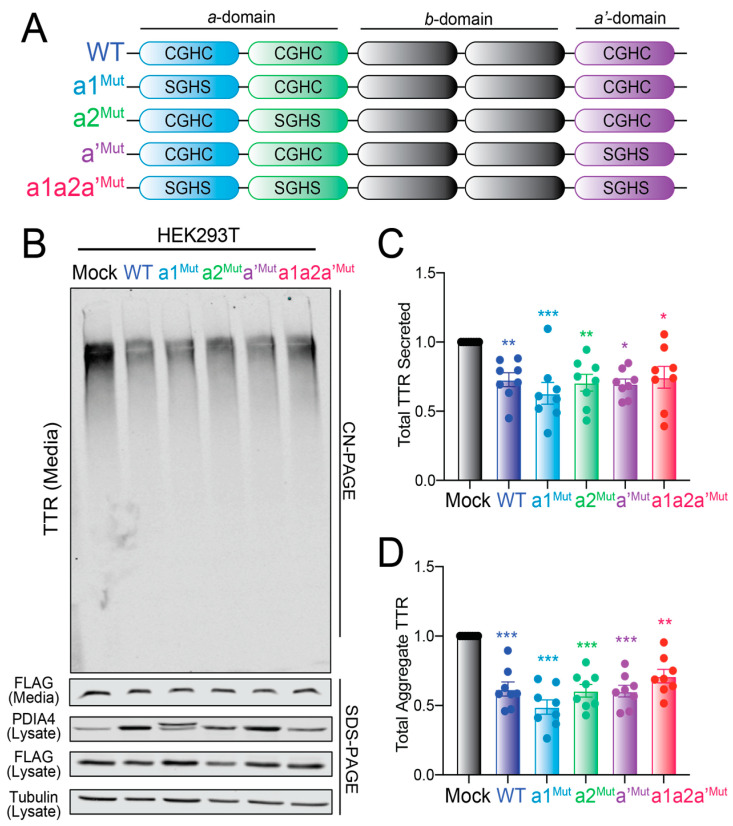
Overexpression of redox inactive PDIA4 variants reduce total and aggregate ^FT^TTR^A25T^ in conditioned media. (**A**) PDIA4 domain architecture showing the three redox active sites of the protein. The specific mutations used in this study are shown. (**B**–**D**) Representative CN-PAGE and SDS-PAGE immunoblot and quantification of total and aggregate ^FT^TTR^A25T^ in conditioned media prepared on transiently transfected HEK293T cells overexpressing mock or the indicated PDIA4 variant (see panel (**A**)). Media were conditioned for 18 h. Error bars show SEM for *n* = 8 independent replicates. * *p* < 0.05, ** *p* < 0.01, *** *p* < 0.005 for an RM one-way ANOVA relative to Mock-transfected cells.

## Data Availability

Not applicable.

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
