# Peer review of "ATF6 Activation Reduces Amyloidogenic Transthyretin Secretion through Increased Interactions with Endoplasmic Reticulum Proteostasis Factors"

_cells, 2022, doi:10.3390/cells11101661_

Round 1
Reviewer 1 Report
The authors identified a putative redox-independent chaperone function of PDIA4 for amyloidogenic TTR. They could show that PDIA4 was upregulated in cells expressing mutated TTR. Co-IP analyses showed an interaction, although it is still unclear if PDIA4 directly binds to TTR or if they are art of a protein complex and PDIA4 would then represent and indirect interaction partner. Nevertheless, overexpression of PDIA4 reduced the secretion of amlyoidogenic but not wt TTR. Surprisingly, the effect of PDIA4 was redox-independent.
Overall, the presented data are sound and of good quality. This study could be strengthened by an in vitro assay showing the interaction of PDIA4 and mutated TTR to address the open question if the effect of PDIA4 is direct or not. What is known about PDIA4? Does ist cooperate with other chaperones?
I also miss a statement by the authors if knockout lines for PDIA4 are available to assess the role of PDIA4 not only by an overexpression system and to also complement and strengthen the data set. An siRNA approach could also be used in case the ko is lethal.
Author Response
Reviewer #1
Reviewer #1 General Comments: “The authors identified a putative redox-independent chaperone function of PDIA4 for amyloidogenic TTR. They could show that PDIA4 was upregulated in cells expressing mutated TTR. Co-IP analyses showed an interaction, although it is still unclear if PDIA4 directly binds to TTR or if they are art of a protein complex and PDIA4 would then represent and indirect interaction partner. Nevertheless, overexpression of PDIA4 reduced the secretion of amlyoidogenic but not wt TTR. Surprisingly, the effect of PDIA4 was redox-independent. Overall, the presented data are sound and of good quality.”
Our Response to Reviewer #1 General Comments. We thank the reviewer for the positive response to our manuscript. We address all of the reviewer’s remaining comments in the revised manuscript as below.
Reviewer #1 Comment #1. “This study could be strengthened by an in vitro assay showing the interaction of PDIA4 and mutated TTR to address the open question if the effect of PDIA4 is direct or not. What is known about PDIA4? Does ist cooperate with other chaperones?”
Our Response to Reviewer #1 Comment #1. The reviewer is specifically asking whether PDIA4 directly binds to destabilized TTR or if it binds as part of a complex. PDIA4 was previously shown to assemble into a complex comprising multiple other ER proteostasis factors including BiP. Due to the short revision timeframe requested by the editor (10 days), we were unable to purify recombinant PDIA4 and TTR to determine whether these two proteins directly interact. However, to determine whether PDIA4 interactions with FTTTRA25T occur independent of BiP binding, we monitored the recovery of BiP in FTTTRA25T immunopurification from cells overexpressing PDIA4. These experiments show that overexpression of PDIA4 increases recovery of PDIA4 in FTTTRA25T IPs by ~100-fold, while only modestly impacting recovery of BiP (Fig. S4A,B). While this does not directly demonstrate direct binding between PDIA4 and destabilized TTR, it does indicate that interactions between destabilized TTR and PDIA4 are independent of the interaction between TTR and BiP. Further, we demonstrate that increases in PDIA4 reduce amyloidogenic TTR secretion in both HEK293 and liver-derived HepG2 cells, while increases in BiP only reduce secretion in HEK293 cells, further highlighting a critical role for PDIA4 levels in defining ER quality control for this disease-relevant protein. We explicitly discuss these new results and the implications of these results in line 405-419 and line 560-573 of the revised manuscript.
Reviewer #1 Comment #2. “I also miss a statement by the authors if knockout lines for PDIA4 are available to assess the role of PDIA4 not only by an overexpression system and to also complement and strengthen the data set. An siRNA approach could also be used in case the ko is lethal.”
Our Response to Reviewer #1. We previously showed that PDIA4 depletion with shRNA causes ER stress and activates the UPR. This precludes our ability to define how reductions in PDIA4 impact TTR secretion independent of the confounding effects of ER stress and UPR activation. This is addressed in our manuscript on line 331.
Reviewer 2 Report
Comments:
This article investigated the molecular basis for ATF6-dependent reductions in amyloidogenic Transthyretin (TTR) secretion and aggregation using SILAC-based quantitative proteomics. The authors found that ATF6 activation reduces amyloidogenic TTR secretion and aggregation through ER retention mediated by BiP and PDIA4. BiP is an already known molecule involved in ER retention of destabilized TTR. The authors reevaluated the function of BiP using their original assays. The originality of this work is that PDIA4 engages in ER retention of destabilized TTR. Interestingly, ER retention of TTR by PDIA4 depends not on the redox activity of PDIA4 but instead on its chaperone activity. These findings will help develop strategies to reduce pathological TTR aggregation by reducing its secretion. However, a significant weakness of this study is that it is unclear whether PDIA4 interacts directly with TTR and collaborates with BiP during ER retention of TTR. ATF6 activation increases both BiP and PDIA4 expression, allowing them to interact with destabilized TTR. This reviewer understands that the difference in the effects of BiP and PDIA4 on TTR secretion depends on the cellular environments, such as basal quality control activity in the ER. Still, mechanistic insight into ER retention of TTR by both chaperones is essential. This reviewer firmly believes that the authors should provide evidence that BiP and PDIA4 work collaboratively or independently of each other during ER retention of TTR.
Minor points:
Figure 1
In figure legends, the labeling is wrong. In line 287, ‘F’ is probably ‘E’. In line291, ‘G’ is perhaps ‘F’.
Figure 3
The authors claim that the reduction in FTTTRA25T secretion from HepG2 cells corresponds with increases in lysate FTTTRA25T. However, the increases in lysate FTTTRA25T by PDIA4 is weak the eye can't even see them. The author should present quantitative data. In addition, unlike HepG2 cells, the increases in lysate FTTTRA25T due to ER retention have not been observed in HEK293 cells. Why?
Figure 6
The authors demonstrated using redox-deficient PDIA4 variants that the reductions in FTTTRA25T secretion afforded by PDIA4 overexpression are attributed to a redox-independent chaperoning mechanism. If PDIA4 directly acts on ER retention of TTR through chaperone activity, then deletion of the PDIA4 ER retention signal (KEEL) would disable TTR secretion inhibition, resulting in both TTR and PDIA4 being co-secreted. This reviewer believes that this experiment serves to strengthen the authors’ argument.
Author Response
Reviewer #2.
Reviewer #2 General Comments. “This article investigated the molecular basis for ATF6-dependent reductions in amyloidogenic Transthyretin (TTR) secretion and aggregation using SILAC-based quantitative proteomics. The authors found that ATF6 activation reduces amyloidogenic TTR secretion and aggregation through ER retention mediated by BiP and PDIA4. BiP is an already known molecule involved in ER retention of destabilized TTR. The authors reevaluated the function of BiP using their original assays. The originality of this work is that PDIA4 engages in ER retention of destabilized TTR. Interestingly, ER retention of TTR by PDIA4 depends not on the redox activity of PDIA4 but instead on its chaperone activity. These findings will help develop strategies to reduce pathological TTR aggregation by reducing its secretion.”
Our Response to Reviewer #2 General Comments. We thank the reviewer for the positive response to our manuscript. We address the reviewer’s remaining comments in the revised manuscript, as below.
Reviewer #2 Comment #1. “However, a significant weakness of this study is that it is unclear whether PDIA4 interacts directly with TTR and collaborates with BiP during ER retention of TTR. ATF6 activation increases both BiP and PDIA4 expression, allowing them to interact with destabilized TTR. This reviewer understands that the difference in the effects of BiP and PDIA4 on TTR secretion depends on the cellular environments, such as basal quality control activity in the ER. Still, mechanistic insight into ER retention of TTR by both chaperones is essential. This reviewer firmly believes that the authors should provide evidence that BiP and PDIA4 work collaboratively or independently of each other during ER retention of TTR.”
Our Response to Reviewer #2 Comment #1 The reviewer is specifically asking whether PDIA4 directly binds to destabilized TTR or if it binds as part of a complex. PDIA4 was previously shown to assemble into a complex comprising multiple other ER proteostasis factors including BiP. Due to the short revision timeframe requested by the editor (10 days), we were unable to purify recombinant PDIA4 and TTR to determine whether these two proteins directly interact. However, to determine whether PDIA4 interactions with FTTTRA25T occur independent of BiP binding, we monitored the recovery of BiP in FTTTRA25T immunopurification from cells overexpressing PDIA4. These experiments show that overexpression of PDIA4 increases recovery of PDIA4 in FTTTRA25T IPs by ~100-fold, while only modestly impacting recovery of BiP (Fig. S4A,B). While this does not directly demonstrate direct binding between PDIA4 and destabilized TTR, it does indicate that interactions between destabilized TTR and PDIA4 are independent of the interaction between TTR and BiP. Further, we demonstrate that increases in PDIA4 reduce amyloidogenic TTR secretion in both HEK293 and liver-derived HepG2 cells, while increases in BiP only reduce secretion in HEK293 cells, further highlighting a critical role for PDIA4 levels in defining ER quality control for this disease-relevant protein. We explicitly discuss these new results and the implications of these results in line 405-419 and line 560-573 of the revised manuscript.
In addition, we further probe the specific potential for overexpressing PDIA4 or BiP to recapitulate the reduction in destabilized TTR secretion observed in cells overexpressing ATF6. Towards that aim, we show that stress-independent activation of ATF6 further reduces amyloidogenic TTR secretion in cells overexpressing PDIA4 or BiP (Fig. S3C). These results demonstrate the predicted integration of multiple ATF6-regulated ER proteostasis factors in remodeling ER quality control and reducing amyloidogenic TTR secretion. These results are discussed in line 353-360 and line 566-573 of the revised manuscript.
Reviewer #2 Comment #2. “Figure 1: In figure legends, the labeling is wrong. In line 287, ‘F’ is probably ‘E’. In line291, ‘G’ is perhaps ‘F’.”
Our Response to Reviewer #2 Comment #2. This is correct. We have fixed the panel labeling in the legend of Figure 1.
Reviewer #2 Comment #3. “Figure 3: The authors claim that the reduction in FTTTRA25T secretion from HepG2 cells corresponds with increases in lysate FTTTRA25T. However, the increases in lysate FTTTRA25T by PDIA4 is weak the eye can't even see them. The author should present quantitative data. In addition, unlike HepG2 cells, the increases in lysate FTTTRA25T due to ER retention have not been observed in HEK293 cells. Why?”
Our Response to Reviewer #2 Comment #3. In Fig. S5C, we show that PDIA4 overexpression increases lysate levels of FTTTRA25T in HEK293 cells. We now also include quantification showing the increase in lysate FTTTRA25T in HepG2 cells overexpressing PDIA4 (see Fig. S3D).
Reviewer #2 Comment #4. “Figure 6: The authors demonstrated using redox-deficient PDIA4 variants that the reductions in FTTTRA25T secretion afforded by PDIA4 overexpression are attributed to a redox-independent chaperoning mechanism. If PDIA4 directly acts on ER retention of TTR through chaperone activity, then deletion of the PDIA4 ER retention signal (KEEL) would disable TTR secretion inhibition, resulting in both TTR and PDIA4 being co-secreted. This reviewer believes that this experiment serves to strengthen the authors’ argument.”
Our Response to Reviewer #2 Comment #4. This is an interesting experiment and a good suggestion. However, due to the short 10 day timeline assigned by the editor for our revised submission, we were unable to perform this experiment for the revised manuscript. Regardless, we feel that the data provided in the revised manuscript are sufficient to reveal an important role for PDIA4 in regulating amyloidogenic TTR secretion.
Round 2
Reviewer 1 Report
The authors have addressed both questions in a satisfactory manner. The manuscript can now be accepted.
Reviewer 2 Report
The authors have responded appropriately to my concerns, providing additional data. I support the publication of the manuscript in the revised form.